# Facet Connectivity-Based Estimation Algorithm for Manufacturability of Supportless Parts Fabricated via LPBF

**DOI:** 10.3390/ma16031039

**Published:** 2023-01-24

**Authors:** Seung-Yeop Lee, Jae-Wook Lee, Min-Seok Yang, Da-Hye Kim, Hyun-Gug Jung, Dae-Cheol Ko, Kun-Woo Kim

**Affiliations:** 1Smart Manufacturing Technology R&D Group, Korea Institute of Industrial Technology (KITECH), Daegu 42994, Republic of Korea; 2R&D Center, STACO Co., Ltd., Ansan-si 15433, Republic of Korea; 3Department of Nanomechatronics Engineering, Pusan National University, Busan 46241, Republic of Korea

**Keywords:** additive manufacturing, manufacturability, geometric analysis, laser powder bed fusion (LPBF)

## Abstract

Recent advances in additive manufacturing have provided more freedom in the design of metal parts; hence, the prototyping of fluid machines featuring extremely complex geometries has been investigated extensively. The fabrication of fluid machines via additive manufacturing requires significant attention to part stability; however, studies that predict regions with a high risk of collapse are few. Therefore, a novel algorithm that can detect collapse regions precisely is proposed herein. The algorithm reflects the support span over the faceted surface via propagation and invalidates overestimated collapse regions based on the overhang angle. A heat exchanger model with an extremely complex internal space is adopted to validate the algorithm. Three samples from the model are extracted and their prototypes are fabricated via laser powder bed fusion. The results yielded by the fabricated samples and algorithm with respect to the sample domain are compared. Regions of visible collapse identified on the surface of the fabricated samples are predicted precisely by the algorithm. Thus, the supporting span reflected by the algorithm provides an extremely precise prediction of collapse.

## 1. Introduction

Additive manufacturing provides considerable freedom in the fabrication materials with diverse geometries. In particular, the fabrication of geometries that include complex internal spaces, e.g., fluid machines such as heat exchangers, has been attempted using additive manufacturing technology [1]. In this study, laser powder bed fusion (LPBF), which is one of the most extensively used methods, is considered as a representative manufacturing technique. This fabrication method involves using a laser beam to fuse metal powder over a bed. The laser beam draws the cross-section profile of a part along the build direction [2]. In this process, a thin layer of metal is laminated over the previous additive part; thus, the geometry of the part is a significant factor in fabrication, despite the numerous environmental parameters that determine the success of fabrication in additive manufacturing. In particular, the overhang angle of the target directly affects the surface roughness [3,4,5]. The overhang angle is the angle between the part surface and the bed plane. When the overhang angle of a part surface is extremely small, the quality of the fabricated part will be unsatisfactory. Figure 1 illustrates the overhang angle and support of a part fabricated via additive manufacturing. A small overhang angle may disrupt the part geometry owing to the gravitational load over the structure or the over-melting of regions during fabrication [6]. Specifically, down skin surface with stiff overhang angle is fused without a solid base in the fabrication process. Thus, support structures are typically implemented on the stiff overhang surface to prevent failures. These support structures not only function as a structural support against gravity to reduce the risk of geometry sinking [7], but also provide a thermal path to dissipate a high concentration of thermal energy, which induces over-melting [8,9,10]. After a part is completely fabricated, these support structures are removed. However, if support structures are implemented in the enclosed internal space of parts with complex geometries, such as heat exchangers, then the removal of the auxiliary structure will be extremely challenging. In addition, if the support structure is not eliminated in the internal space of the heat exchanger, the residual support structure will adversely affect the flow and heat transfer, thus resulting in undesirable responses from the heat exchanger. Therefore, the geometry of parts must be modified to restrain support generation in undesirable internal spaces. To modify the part geometry for a more stable fabrication, the area requiring support must be identified. Meanwhile, collapse detection requires the consideration of thermal and structural aspects [11]. Although thermal and structural attributes should be addressed to precisely detect collapsed regions, overestimated support regions determined based on the overhang angle have been extensively adopted to reduce the simulation time. Overhang angle-based estimations provide high efficiency in a timely manner; however, the precise detection of collapse regions is necessary to reduce the material cost or minimize revision for the supportless design of internal spaces wherein the support structure cannot be eliminated [12]. Recently, numerous studies have been conducted on the quality and characteristics of the parts fabricated by additive manufacturing [13,14,15]. However, studies that consider the role of geometry in precisely determining support regions are few. Wang et al. [16] presented a methodology that involved a detection algorithm to optimize support points on a part, and Huang et al. [17] proposed a methodology that involved using a convolutional neural network to detect regions requiring support. To classify the actual support region, not only the overhang angle, but also a geometrical analysis must be considered. Exceeding the critical overhang angle does not necessarily induce collapse. As demonstrated experimentally by Vora et al. [18], the fabrication of an overhanging part using a powder bed fusion method, i.e., anchorless selective laser melting (ASLM), yielded a partially overhanging part, although the fabrication was unsuccessful.

This indicates that an overhanging part in the vicinity of the support is partially supported by its geometry, although the overhang angle of the part is much smaller than the critical angle. Figure 2 shows an example of an unsuccessful fabrication and a residual overhanging structure. Based on the remaining part, an overhanging part is supported by an adjacent stable structure around the boundary edge. The effect of support must be considered when determining the actual collapse region.

Target geometries for additive manufacturing must be verified in advance to minimize material and time losses. Although the precise prediction of collapse regions based on the overhang angle is critical, it has been addressed in only a few studies. Thus, an algorithm is developed in this study to estimate the actual probability of a surface requiring reinforcement by support structures. The algorithm identifies the actual collapse region by reflecting the geometric attributes of the support and overhanging part by the invalidation of the overestimated collapse region. Figure 3 shows the supporting boundary edge (blue line), the actual collapse region, and the supporting span (red line). The spanwise supporting area is determined along the supporting boundary edge of the conformal span, as shown in Figure 3. In this study, the spanwise supporting area is determined by estimating the overhang angle, and the algorithm re-estimates the collapse region over the part surface. In fact, the algorithm can be used for any complex part, including those comprising extremely complicated boundary edges and surfaces.

## 2. Algorithm for Supporting Effect

### 2.1. Facet Orientation Angle Estimation

To investigate the support effects, the orientation angle over the part surface should first be estimated. Hence, a geometrical analysis was performed using STL (standard tessellation language) formats. The STL format includes three-point facet surface data and their normal vectors. The geometry of the parts is described by a set of facets and normal vectors toward the external directions of the facets. Figure 4 illustrates a facet part in the build space. The orientation angle of the facet is determined as follows:(1)$$\theta_{i}=cos^{-1}\left(\frac{-\hat{k}\cdot {\hat{n}}_{i}}{\left|-\hat{k}\right|\left|{\hat{n}}_{i}\right|}\right)=cos^{-1}\left(-\hat{k}\cdot {\hat{n}}_{i}\right)$$
where ${\hat{n}}_{i}$ is the normal vector of the facet *i* and $-\hat{k}$ is the negative unit vector in the build direction. The index of the facet is denoted by subscript *i*. The estimated orientation angle provides a broad spectrum of collapse region detection that includes overestimated collapse regions. The orientation angle-based collapse region detection assumes any surface oriented with an orientation angle below a critical overhang angle is a collapse region. The critical overhang angle is generally defined based on experimental results while considering the surface roughness and part disruption [19]. The critical overhang angle can be evaluated accordingly based on the individual tolerance of the functional attributes of the target part. In addition, the critical overhang angle is affected by numerous fabrication factors such as material and build parameters, e.g., scan speed and laser power. Thus, the orientation angles on the part surface and the quality of the fabricated part must be evaluated to determine the suitable critical overhang angles for various fabrication conditions and applications. Yang et al. [20] investigated the effect of orientation angle in LPBF, as it determines the surface quality of parts fabricated using AlSi10Mg metal powder.

### 2.2. Initial Step for Assigning Initial Value on Supporting Boundary Edge

The algorithm reflects the support span on a set of facet data, and the geometric attributes of the overhanging part are recognized as connections among facets by their edges. As the initial procedure of the algorithm, edge searching over facets is performed to determine the supporting boundary edge as a set of supporting edges for each facet. Any facet part of a solid in STL format is connected with three adjacent facets. The supporting edges are evaluated within the facet connections based on the decision process. Figure 5 presents a decision tree to search for a supporting edge. The search procedure is executed over all three adjacent facets in the collapse region, which are oriented at an angle less than the critical angle ($\theta_{c}$). The supporting edge only exists when a facet within the collapse region is supported by an adjacent stable facet whose orientation angle exceeds the critical overhang angle. Thus, the orientation angles of the three adjacent facets should be addressed to classify the supporting edges.

To detect supporting edges in a pair of stable facets and a facet collapse region, the orientation angles of three adjacent facets are first addressed. Subsequently, qualified edges are determined only between a facet with a smaller orientation angle than the critical overhang angle and a facet with an orientation angle larger than the critical angle. If any of the part facets with a larger critical overhang angle are constructed earlier than the supporting facet, then the support effect will not be transferred to the unstable facet because additive manufacturing is a sequential procedure along the build direction. Therefore, the previously qualified edges are evaluated via the build sequence constriction. The build orders of a facet and its adjacent facet are classified via a comparison of their points.

Figure 6 shows the connection between a facet and its adjacent facet. Each facet comprises two shared points and one free point because two facets share a common edge. Hence, the build order is determined by comparing the *z*-coordinate of the free point of each facet. If the *z*-coordinate of the free point of the adjacent facet ($z_{{\left.i_{ad}\right|}_{j}}$) is above that of the corresponding facet ($z_{{\left.i\right|}_{j}}$), then the adjacent facet is constructed after the corresponding facet is built. The supporting edges are determined after considering the build order.

An example of supporting edge detection is presented in Figure 7. A facet *i* with orientation angle zero and three adjacent facets is illustrated. The example shows the methods to determine the supporting edge among three edges of facet *i* when the critical overhang angle is 45°. Details regarding the supporting edge detection are presented as follows:1.The shared edge between facet i and adjacent facet 1The orientation angle of facet *i*, $\theta_{i}$, is smaller than the critical overhang angle, 45° (first qualification);The orientation angle of adjacent facet 1, $\theta_{{\left.i_{ad}\right|}_{1}}$, is larger than 45° (second qualification);The z-coordinate of a free point of facet *i* with adjacent facet 1, $z_{{\left.i\right|}_{1}}$, is below the free point of adjacent facet 1, $z_{{\left.i_{ad}\right|}_{1}}$ (third qualification).

The dismissed edge 1 between facet *i* and adjacent facet 1 is disqualified by the third qualification.

2.The shared edge between facet i and adjacent facet 2The orientation angle of facet *i*, $\theta_{i}$, is smaller than the critical overhang angle, 45° (first qualification);The orientation angle of adjacent facet 2, $\theta_{{\left.i_{ad}\right|}_{2}}$, is smaller than 45° (second qualification).

The dismissed edge 2 between facet *i* and adjacent facet 2 is disqualified by the second qualification.

3.The shared edge between facet i and adjacent facet 3The orientation angle of facet *i*, $\theta_{i}$, is smaller than the critical overhang angle, 45° (first qualification);The orientation angle of adjacent facet 3, $\theta_{{\left.i_{ad}\right|}_{3}}$, is larger than 45° (second qualification);The z-coordinate of a free point of facet *i* with adjacent facet 3, $z_{{\left.i\right|}_{3}}$, is above the free point of adjacent facet 3, $z_{{\left.i_{ad}\right|}_{3}}$ (third qualification).

Thus, the shared edge 3 between facet *i* and adjacent facet 3 is classified as the supporting edge.

To reflect the spanwise supporting area along the detected supporting edges, the initial value of the supporting span should be decided based on the facet collapse region, which includes the detected edges. The initial value along the supporting edges is defined as follows:(2)$$\Phi_{i}=S_{i}-\alpha \sum l_{{\left.i\right|}_{s}}$$
where $\Phi_{i}$ is the initial value assigned to the corresponding facet *i* (instead of its area $S_{i}$), $S_{i}$ and $l_{{\left.i\right|}_{s}}$ represent the facet area and the length of the supporting edge (*s*) of a facet with index *i*, respectively. If a facet consists of a supporting edge, the initial value $\Phi_{i}$ is assigned with a negative value in the initial step owing to the larger rectangular area yielded by the multiplication of α and $\;l_{{\left.i\right|}_{s}}$. However, a surface area is assigned to the facet if all edges of the facet are dismissed. The initial value setup based on a modeling example is illustrated in Figure 8. The coefficient α, which represents an effective length, must be considered when assigning the supporting distance from the boundary edges, and the facets are assigned by negative value of $\Phi_{i}$, because the area of the rectangle with effective length α and supporting edge length $l_{{\left.i\right|}_{s}}$ is larger than the area of facet, $S_{i}$. An initial value $\Phi_{i}$ must be assigned over the initial facets along the supporting boundary edges for further analysis of the presented algorithm.

### 2.3. Edge Classification

Spanwise supporting is addressed over the surface with facets via an iterative propagation from the initial values on the facets along the supporting boundary edge. However, the facet to be supported by adjacent facets must be identified. In this regard, a decision procedure to categorize edges is proposed. The support effect is propagated from one facet to another; thus, the edge category is determined by considering the interconnection of a pair of facets. Although the facets share an identical edge, the edge type can be evaluated differently depending on the facet from which the propagation originated. Figure 9 illustrates the decision tree used to categorize an edge into a propagative edge (PE) or a non-propagative edge (NPE). This categorization determines whether the effect is transferred to the other facet across the corresponding edge. First, the orientation angles of the adjacent facets of a facet from which propagation originated should be considered. If an adjacent facet features an orientation angle larger than the critical angle, then the origin facet does not transfer the support effect to the adjacent facet across the common edge of the two facets. Next, the sequential order of two facets is considered to classify the edge. The origin facet only propagates to an adjacent facet with a later build order because the support effect is only propagated from the pre-built facet to the post-built facet owing to the build sequence of the LPBF method. Propagation supervision is critical, not the transfer of the effect in a backward direction. Thus, the origin facet does not transfer the effect to an adjacent facet whose propagation effect has been calculated.

### 2.4. First Step for Transferring Support Effect λ

The first step of the propagation process begins from the initial facets assigned with initial values. The values of the transferring effect are defined by an assigned value, the edge category, and length of the edges of the origin facet. The transferring value is calculated as follows:(3)$$\lambda_{{\left.i\right|}_{j}}=\min\left(\Phi_{i},0\right)f_{{\left.i\right|}_{j}}g_{{\left.i\right|}_{j}}$$
where $\lambda_{{\left.i\right|}_{j}}$ is the value of the support effect and *i*, and *j* are the facet index and index of edges in the facet, respectively. The initial value of propagation is assigned a negative value; therefore, the effect is propagated only when a facet is assigned a negative value. The auxiliary variables $f_{{\left.i\right|}_{j}}$ and $g_{{\left.i\right|}_{j}}$ are defined as follows:(4)fij=1       if j of facet i is PE 0    if j of facet i is NPE
(5)$$g_{{\left.i\right|}_{j}}=\left\{\begin{array}{cc} 0 & \overset{3}{\underset{j=1}{\sum }}\left(f_{{\left.i\right|}_{j}}l_{{\left.i\right|}_{j}}\right)=0\\ l_{{\left.i\right|}_{j}}{\left\{\overset{3}{\underset{j=1}{\sum }}\left(f_{{\left.i\right|}_{j}}l_{{\left.i\right|}_{j}}\right)\right\}}^{-1} & \overset{3}{\underset{j=1}{\sum }}\left(f_{{\left.i\right|}_{j}}l_{{\left.i\right|}_{j}}\right)\neq 0 \end{array}\right.$$

When the edge *j* is an NPE, the transferred value, λ, is always zero. The function $g_{{\left.i\right|}_{j}}$ involves the reciprocal of the sum of all lengths of the PE because λ is transferred based on the proportion of the length of the corresponding PE *j* to the total length of all PEs. Figure 10 illustrates the first step of the propagation process. The support facets on the surface are shown in red, and the facets assigned with initial values are shown in blue. At facet *i*, i.e., the origin facet of propagation, the propagation of λ shows that λ is transferred differently depending on the edge category. The third edge of facet *i* is shared by a facet on the support structure and a facet in the collapse region; thus, the third edge is categorized as an NPE, which always transfers a zero value. Meanwhile, the first and second edges are shared by adjacent facets on the parallel collapse region surface, and the adjacent facets are not calculated in the previous iterations. Hence, the two edges satisfy all conditions of a PE, which transfers values across the edge.

### 2.5. Second Step for Receiving Support Effect λ

In the second step of the propagation process of the algorithm, the newest value Φ is assigned to adjacent facets that have not been calculated in previous iterations. The newest assigned value is determined from three adjacent facets and is expressed as follows:(6)$$\Phi_{i}\leftarrow \Phi_{i}+\overset{3}{\underset{j=1}{\sum }}\lambda_{{\left.i_{ad}\right|}_{j}}$$
where the newest value Φ is updated by accumulating all values of λ from the adjacent facets. The number indexes of the edges are reorganized based on the affected facet *i*; therefore, the number indexes of the edges are not identical with the number indexes of the origin facet in the first step of propagation, although the affected facet and origin facet share an identical edge. Thus, the transferred value, $\lambda_{{\left.i_{ad}\right|}_{j}}$, is denoted as a value from an adjacent facet sharing edge *j* with a facet *i*, not as its own index. This second step is the final sequence of the single iteration for propagation. The first and second steps are iterated. However, the negative value of Φ approaches zero as the analysis proceeds over a larger area because the magnitude of the negative value is diminished by the newest facet area, as indicated by Equation (2) for the second step. When Φ becomes a positive value, the iterative analysis is completed, because the value of λ is no longer defined as shown in Equation (3) from the first step. Figure 11 illustrates the second step, where λ values are received from two adjacent facets, Φ is defined as the further propagation, and indexes of the facets and edges are numbered to aid discussion. The assigned value $\Phi_{3}$ is determined by two λ values obtained using Equation (6). However, the λ value transferred across the edge 3 of facet 3 is defined as zero because the adjacent facet sharing the third edge is undefined, although the λ values across the edge 1 and edge 2 of facet 3 are defined.

### 2.6. Process Demonstration

A calculation example is presented in this subsection based on specific cases for the first and second steps. Figure 12 illustrates the propagation of the support effect from the index of the number 3 origin facet to the affected facet numbered 8. In the first iteration, the origin facet, indicated in green, transfers λ across two NPEs and one PE. If a negative value is assigned to $\;\Phi_{3}$, then the values of λ transferred across the NPE are zero, and the λ value transferred across the PE is identical to the assigned value $\Phi_{3}$. The specific order to calculate the λ value across the PE is as follows:

1.$\lambda_{{\left.3\right|}_{2}}=\min\left(\Phi_{3},0\right)f_{{\left.3\right|}_{2}}g_{{\left.3\right|}_{2}}$, $\Phi_{3}<0$Based on Equation (3);2.

$f_{{\left.3\right|}_{2}}=1$

Based on Equation (4);3.

$g_{{\left.3\right|}_{2}}=1$

Based on Equation (5);4.$\lambda_{{\left.3\right|}_{2}}=\Phi_{3}$.

In the second iteration, the value assigned to the affected facet 8 with a dashed outline is defined by three λ values from two undefined facets and one previously defined facet. The transferred λ values are zero for edges shared with two undefined facets, and the λ value from the defined facet is $\;\Phi_{3}$. The newest assigned value $\Phi_{8}$ is determined in the second propagation step. The specific order of the calculation is as follows:

1.

$\Phi_{8}=S_{8}+\lambda_{{\left.3\right|}_{2}}$

Based on Equations (2) and (6);2.$\lambda_{{\left.3\right|}_{2}}=\lambda_{{\left.8_{ad}\right|}_{1}}=\Phi_{3}$;3.$\Phi_{8}=S_{8}+\Phi_{3}$.

The procedures above are repeated over the faceted surface until the origin facets in the first step constantly transfer values of zero to their adjacent facets in subsequent iterations. This allows the affected facets to be determined by the supporting span over the surface. Consequently, the latest propagation procedure returns the value Φ over all facets within the support span. The assigned value further invalidates the overestimated support region to detect geometry disruptions precisely.

### 2.7. Invalidation of Overestimated Collapse Region

The support effect based on the span area is reflected by the propagation procedure explained in earlier sections. Based on the result of the propagation, any facet assigned with a negative Φ value can be regarded as a robust facet stably supported by adjacent facets, although the facet is oriented at an orientation angle smaller than the critical overhang angle. Thus, stable facets in the collapse region estimated based on the critical overhang angle can be disregarded to precisely specify the collapse region. The invalidation procedure is performed by re-evaluating the orientation angle over the collapse region. The facet orientation angle is re-evaluated as follows:(7)ϑi=θc                Φi≤0 θi                Φi>0
where $\vartheta_{i}$, $\theta_{c}$, and $\theta_{i}$ denote the re-evaluated orientation angle of the facet *i*, the critical overhang angle, and the orientation angle defined by a normal vector of the facet *i*, respectively. Changing the orientation angle of a facet to the critical overhang angle implies that the risk region is a stable region.

## 3. Experimental Validation via LPBF

The algorithm presented herein was developed to precisely detect collapse regions by predicting the over-melting area. Thus, the algorithm was applied to a target model with an extremely complex enclosed internal space to predict surface collapse in the enclosed space. Several sample sections were extracted based on high-risk areas evaluated via critical overhang angle-based risk estimation. Domains of the extracted samples were fabricated using AlSi7Mg metal powder. The surface disruption of the fabricated samples was compared with the result yielded by the proposed algorithm for verification.

### 3.1. Heat Exchanger Model

The algorithm was applied to a heat exchanger model (the target model) that featured a complicated enclosed space. Moreover, a collapsed internal surface, or one with unsatisfactory quality, induces severe problems, e.g., detached metal fragments, which can flow into the fluid circulation system. Thus, collapse areas must be predicted in advance for applications pertaining to heat exchangers. Figure 13 shows the orientation of a part for a building and heat exchanger, which features an extremely complex enclosed internal geometry. The orientation angle was 69°, which was specified by considering the chamber size and construction duration. The axisymmetric exterior of the target model allows the support structure to be eliminated easily. However, the complex interior of the target model renders it impossible to remove the support structure. Hence, the surface stability of such a model must be primarily addressed based on an internal space. Surface collapse risk was first estimated based on the orientation angle of the internal surface. Figure 14 illustrates a contour of the orientation angle of the part surface. Based on previous studies, an orientation angle ranging from 0° to 45° should be prioritized [21]. An area with an orientation angle of approximately 0° can be regarded as an area with a high risk of collapse. Thus, the following sectors were selected for detailed analysis: (1) sectors comprising a surface with an orientation angle of 0°, which was assumed to be a high-risk region; (2) sectors comprising visible high-risk regions; and (3) sectors comprising large high-risk regions in spaces where the support structure could not be eliminated. Based on the criteria above, three sectors were selected, presented by red boxes in the top view of Figure 14.

Based on the three selected sectors, the corresponding samples were extracted from the target model. Printing the entire target part for the manufacturability test is time-consuming and incurs a high material cost. Thus, fabricating representative samples for testing is preferred [22]. Figure 15 shows the domain and position of each sample. The samples were fabricated at the same orientation as that by which the target model was arranged on the bed plate, such that equivalent attributes of the entire model domain and orientation were reflected.

### 3.2. Fabrication

Prototypes of the extracted samples were fabricated via LPBF for further experimental validation. However, the process parameters of additive manufacturing directly affect the quality of the fabricated part. Thus, the conditions of the additive manufacturing process are specified herein. The material is one of the major parameters affecting additive manufacturing. Hence, the properties of the material were first analyzed. AlSi7Mg metal powder was adopted as the material of the samples. The characteristics of the metal powder grain must be specified; thus, microscopic images of the powder were captured via scanning electron microscopy (SEM). Figure 16 shows the SEM images in various magnifications. The grain size was measured to be 15–63 µm based on the SEM images. The thermal and physical properties of the material were measured experimentally; in particular, the thermal properties were measured at two different temperatures, i.e., 25 °C and 90 °C. The values of measured properties of AlSi7Mg powder are listed in Table 1. Additionally, the operation parameters used in additive manufacturing affect the fabrication process; hence, the process parameters used in the LPBF process are listed in Table 2.

Figure 17 illustrates the laser projection during the LPBF process. A concept laser M2 series 5 was used in the fabrication of samples. The operation parameters are specified by an infill and contours. When the laser drew an infill of the cross-sectional profile of the part, the laser fused grains via 370 W of power at 1300 mm/s of transitional velocity, and the beam diameter was 0.11 mm at a hatch distance of 0.14 mm. When the laser was projected along the path of the profile outline, the laser power and scanning speed were identical to the infill. However, the beam diameter was 0.075 mm, which is relatively small for printing the surface of the part precisely. The layer thickness was 0.06 mm. Figure 18 shows the build orientation of the three samples. The samples were fabricated with an orientation angle of 69° as well as the samples that were orientated in the heat exchanger. The fabricated samples are illustrated in Figure 19. The first, second, and third samples are shown in Figure 19a–c, respectively. The samples for the second and third sample were cut along the dashed red line to obtain upper and lower pieces such that the internal space can be observed, as presented in Figure 19b,c.

## 4. Results and Discussion

The collapse detection algorithm was applied to the extracted sample parts. MATLAB^®^ software, https://ww2.mathworks.cn/products/matlab.html (accessed on 21 December 2022) was adopted to execute all the protocols of the algorithm using a computing interface comprising an Intel^®^ Zeon^®^ E5-2697 v3 (14 cores) dual processor. The correct representation of the complex geometry requires numerous fine facets, which increases the calculation time and thus the processing time of the entire computation process. The number of facet elements to describe each sample and the number of iterations for estimation are listed along with the processing time in Table 3. The collapse over the surface of the samples was visualized macroscopically and then evaluated. The recognizable features of the collapse regions were analyzed individually. Detailed comparative evaluations by sample are presented in the following.

### 4.1. Sample 1

In sample 1, no visible and intolerable collapse regions were observed. To provide a more detailed comparison, four features of the sample were analyzed, as shown in Figure 20. Each feature was named by numbers, and the result estimated by the algorithm was comparatively evaluated with the estimation based on the critical overhang angle; subsequently, the algorithm was validated based on the fabricated samples. Figure 21 shows the contours of the orientation angle over the surface of sample 1, and the overestimated collapse region was invalidated in the contour from the algorithm-based estimation. Visible features on the sample yielded by two different estimation methods were compared based on their features. The critical overhang angle-based estimations of feature1 and feature4 present a surface with a relatively higher orientation angle compared with those of feature2 and feature3, as shown in Figure 20. However, the algorithm-based estimation does not highlight any regions over the sample surface, confirming the stability of the surface of sample 1, because the region with a smaller orientation angle is supported by its adjacent structure with the supporting span. Clearly, the algorithm-based estimation was supported by the result of the fabricated sample, as shown in Figure 20. The fabrication of sample 1 was successfully completed without any collapse over the surface, as indicated by the close-up images of feature1, feature2, feature3, and feature4 (see Figure 20).

### 4.2. Sample 2

Sample 2 was fabricated to verify the manufacturability. In particular, sample 2 was segmented into two pieces to observe the collapse regions over the surface of the internal space. Collapse was clearly detected on the surface of sample 2, as shown in Figure 22. Subsequently, features were selected on the visible collapse regions and named. Feature1 and feature2 denote the observed collapse regions in the upper piece of sample 2, as shown in Figure 22a. Similarly, the lower piece of the sample exhibited visible collapse regions, as shown in Figure 22b. A few collapse features with the same feature name imply an originally singular region that had been segmented via sample cutting for observation. The collapse regions on the upper segmented region of feature1 and feature2 were clearly visible, as shown in Figure 22a. However, the lower segmented region of sample 2 shows relatively less collapse, although the collapse regions over both the upper and lower regions were unacceptably rough and not stable enough to be used in the heat exchanger. Figure 23 illustrates the orientation angle and the re-evaluated orientation angle of sample 2. The critical overhang angle-based estimation shows uniformly distributed risky orientation angles over the regions in the segregated parts. Predicting or specifying collapse regions based on only the contour of the orientation angle is unfeasible. However, the algorithm-based estimation shows various risky orientation angles for the segmented parts. Feature1 on the upper region of sample 2 shows numerous facets with a low orientation angle, whereas the lower region of sample 2 shows a predominantly invalidated region; similar tendencies were indicated for feature2. The contour from the algorithm-based estimation suggests that the collapsed region in the upper region will exhibit more severe surface disruption.

Based on the close-up images of the collapsed regions in sample 2, the algorithm-based estimation is more suitable for predicting collapse on the surface. Figure 24 shows the zoomed-in images of the features in the segmented regions of sample 2. The prediction by the algorithm is more precise compared with the result of the critical overhang angle-based estimation. The disruption of the surface of feature1 in the upper region featured clearly visible surface collapse with an extremely rough surface, as shown in Figure 24a. In contrast, feature1 in the lower region shows less disruption and a lower level of surface roughness, as shown in Figure 24b. Similarly, feature2 shows slight collapse and a lower level of surface roughness in the lower region, whereas feature2 shows clearly visible surface disruption and a coarse surface in the upper region of sample 2, as shown in Figure 24a,b.

### 4.3. Sample 3

To validate the results of the algorithm, sample 3 was fabricated. Figure 25 shows sample 3 segmented into upper and lower regions. Five regions with surface disruption were observed on sample 3 and named. However, collapse regions were only identified in the upper region of sample 3. Figure 26 shows the contours of the orientation angle obtained via the critical overhang angle-based and algorithm-based estimations. The critical overhang angle-based estimation shows a line shape for low orientation angle regions along concave edges. Furthermore, widely distributed low orientation angles were shown over the surface of the upper and lower regions of sample 3. However, the lower region of sample 3 was successfully fabricated without any collapse and rough sections, as shown in Figure 25b. In contrast, the algorithm estimates that low orientation angle risk does not exist on concave edges, as shown in Figure 25a. Meanwhile, the contour of the orientation angle re-evaluated by the algorithm invalidated any low orientation angle risk over the surface of the lower region. Figure 27 shows the five collapsed features on the upper region of sample 3. Feature1 and feature2 were described more precisely by the algorithm-based estimation than by the critical overhang angle-based estimation because the lower region of the segmented sample was fabricated precisely, i.e., in contrast to the prediction of the critical overhang angle-based estimation. Similarly, feature3, feature4, and feature5 were described accurately by the algorithm-based estimation, whereas the critical overhang angle-based estimation indicated overestimated risk along the concave edges.

## 5. Conclusions

Collapse regions must be predicted for applications of fluid machines that involve complex internal surfaces. An algorithm for the enhanced detection of collapse regions was presented herein. The algorithm reflects the supporting span over a faceted STL geometry to specify the actual collapse region over the surface of a part fabricated by additive manufacturing. The algorithm involves edge classification and a propagation process, where the supporting effect is transferred over the geometry surface. The proposed algorithm enables the detection of collapse regions over complicated nonlinear surfaces. In this study, the proposed algorithm was experimentally validated using three different samples. Results showed that the proposed algorithm provided a more comprehensive description of the collapse region profile compared with the critical overhang angle-based estimation. Furthermore, the results indicate that the proposed algorithm is suitable for extremely complex geometries. Thus, the proposed algorithm can be applied to predict the surface quality of parts fabricated via additive manufacturing [23] as well as to solve self-supported structures [24] and build orientation problems [25]. Moreover, multiphysics simulations are typically time consuming; however, using the proposed algorithm enables geometry analysis to be performed numerically in a reasonable amount of time.

## Figures and Tables

**Figure 1 materials-16-01039-f001:**
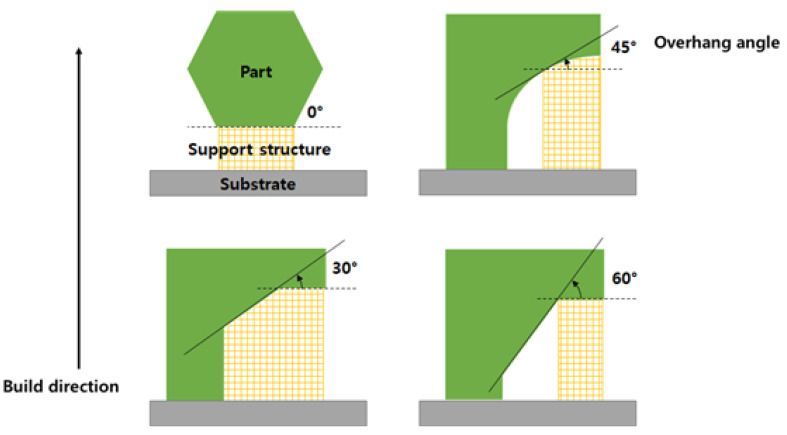
Overhang angle and support of part fabricated via additive manufacturing.

**Figure 2 materials-16-01039-f002:**
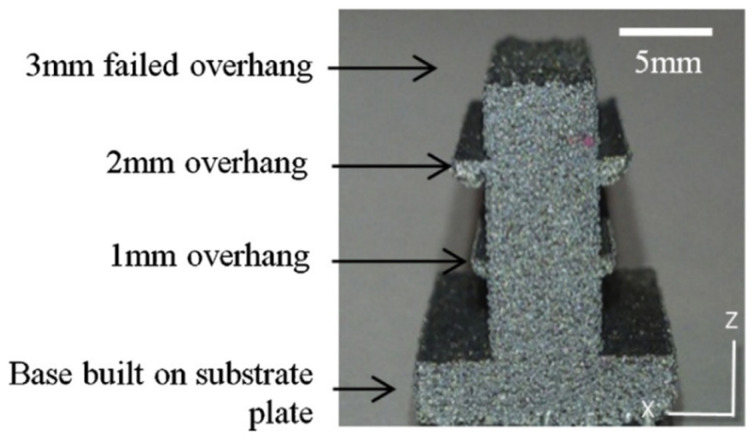
ASLM overhanging part constructed at 100 °C (Vora et al., 2015 [18]).

**Figure 3 materials-16-01039-f003:**
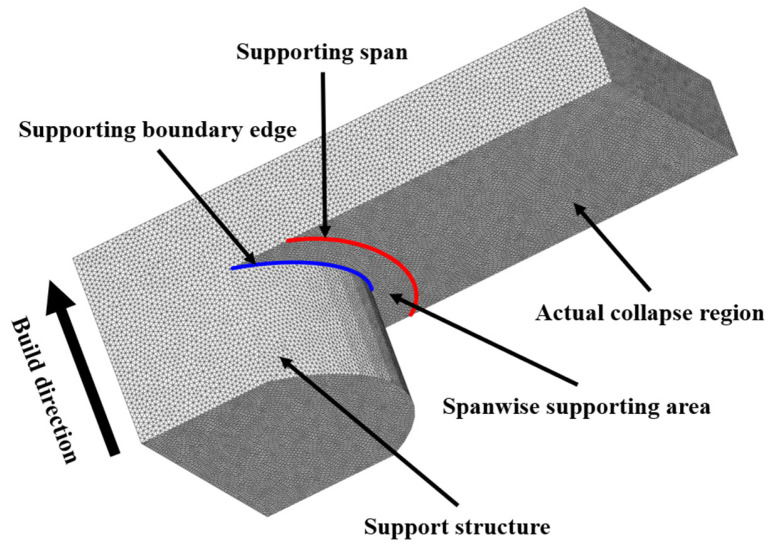
Overhanging structure with spanwise supporting area.

**Figure 4 materials-16-01039-f004:**
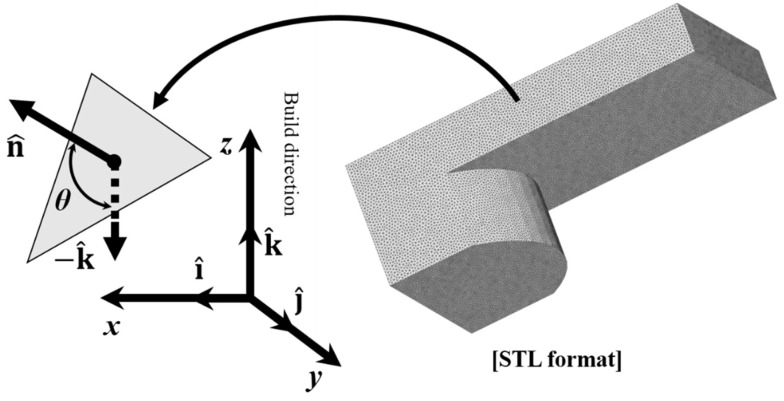
Orientation angle and normal vector of facet in STL format.

**Figure 5 materials-16-01039-f005:**
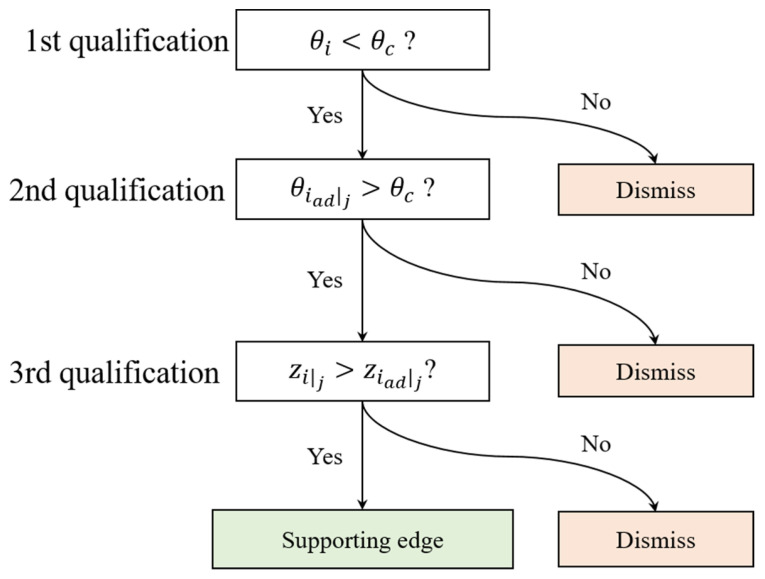
Decision tree for detecting supporting edges within interconnection of facets.

**Figure 6 materials-16-01039-f006:**
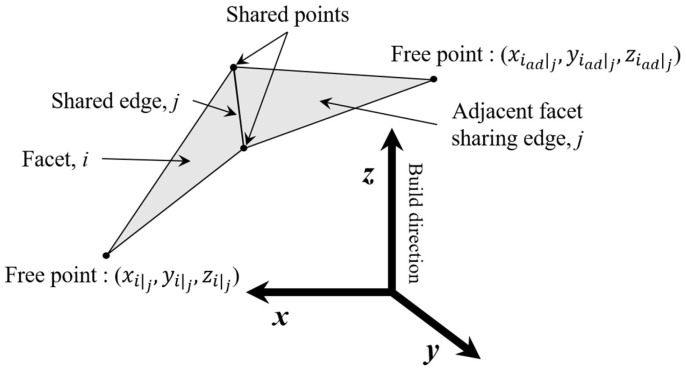
Geometric interconnection of a pair of facets.

**Figure 7 materials-16-01039-f007:**
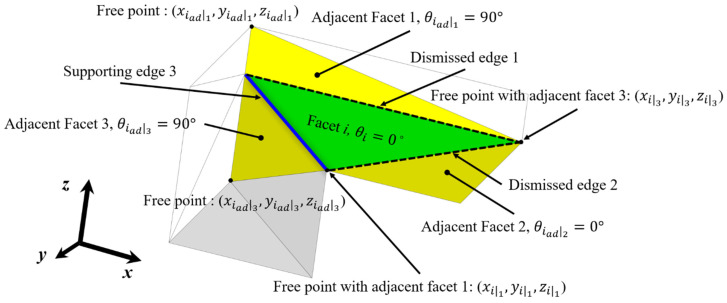
Method to determine the supporting edge.

**Figure 8 materials-16-01039-f008:**
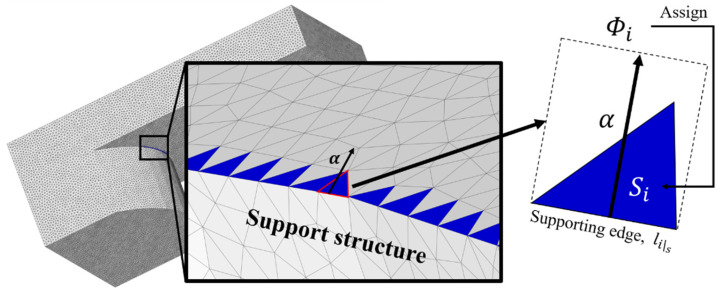
Initial step of collapse region search algorithm proposed herein.

**Figure 9 materials-16-01039-f009:**
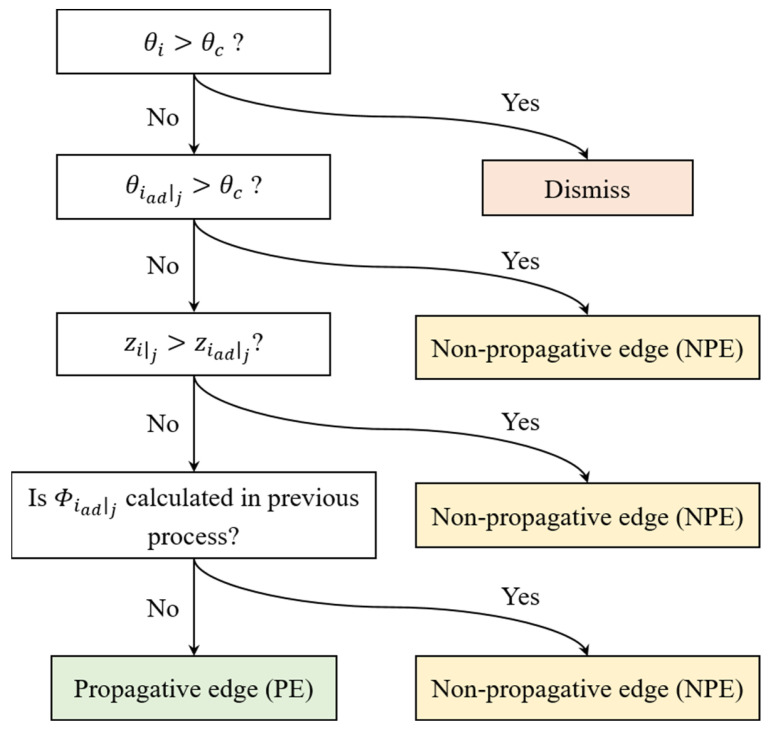
Decision tree for classification of edge *j* of an origin facet based on attributes of an adjacent facet.

**Figure 10 materials-16-01039-f010:**
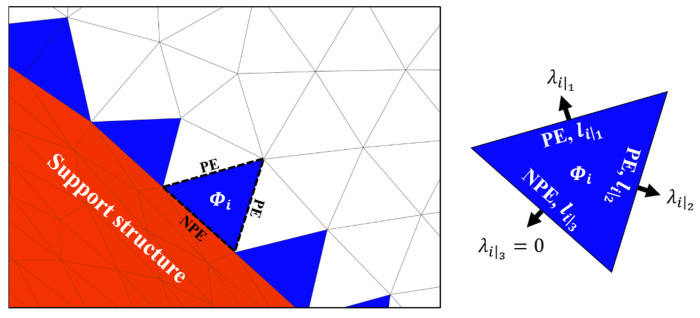
First step of the propagation process for the proposed algorithm.

**Figure 11 materials-16-01039-f011:**
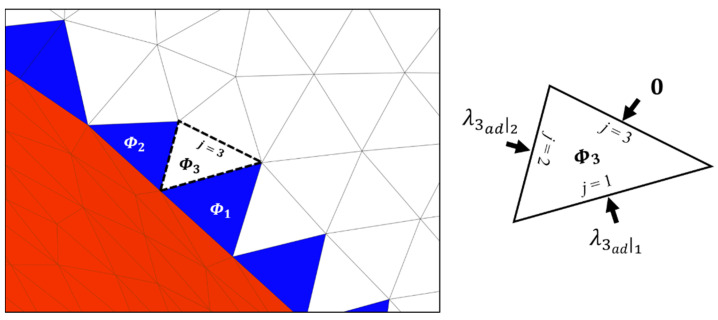
Second step of the propagation procedure of the proposed algorithm.

**Figure 12 materials-16-01039-f012:**
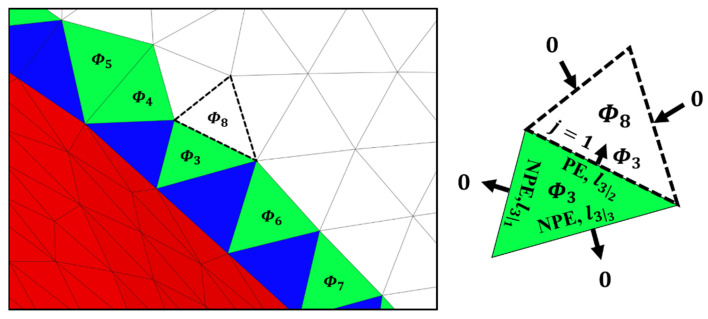
Example case of further propagation.

**Figure 13 materials-16-01039-f013:**
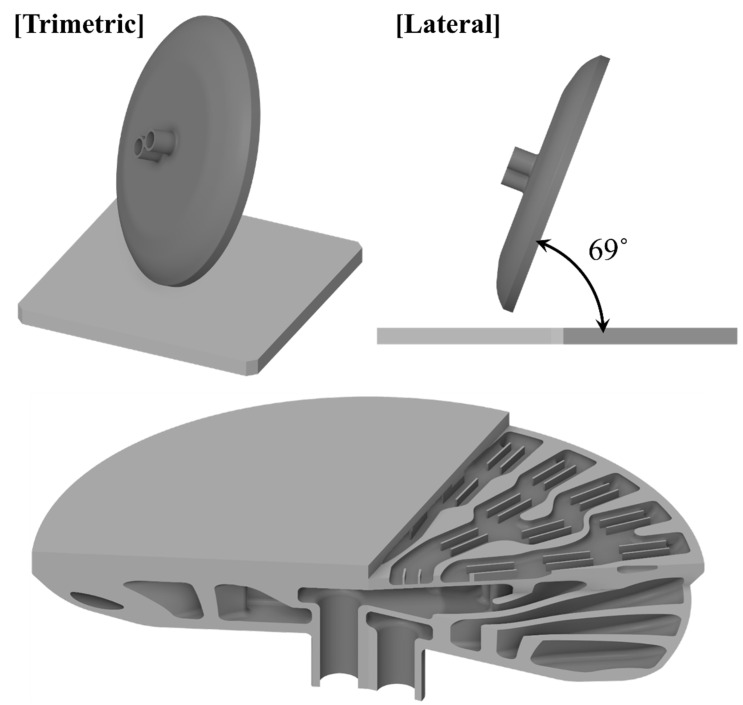
Build orientation and internal space of target heat exchanger model.

**Figure 14 materials-16-01039-f014:**
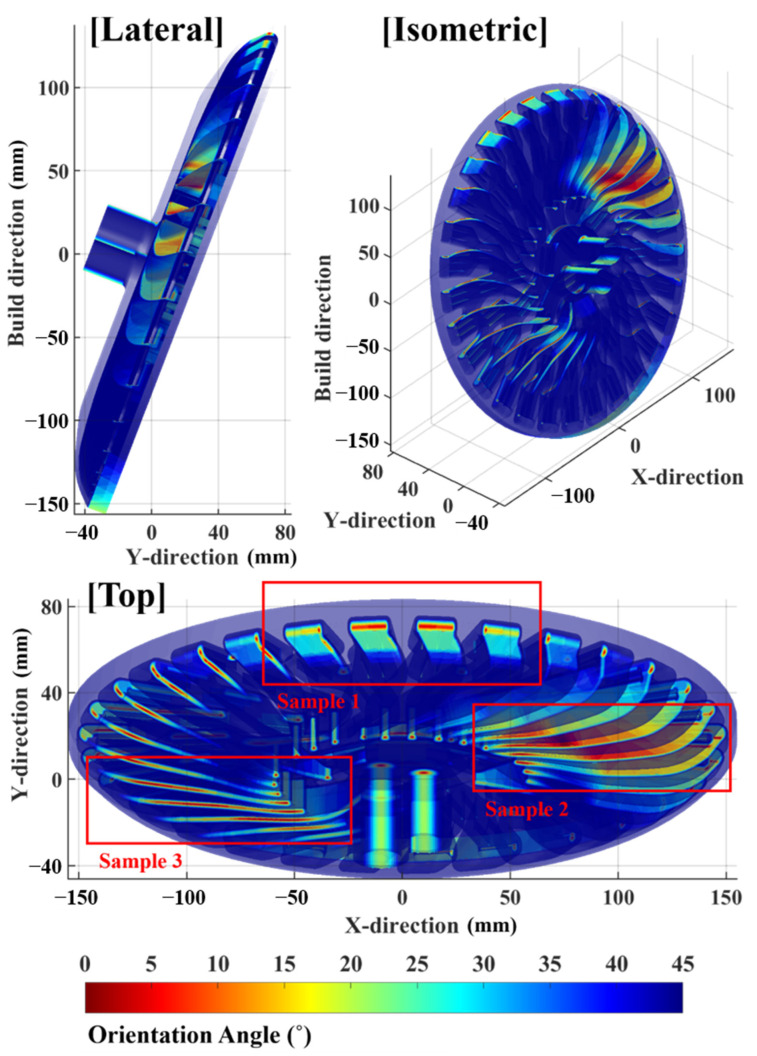
Transparent contour of orientation angle over the surface of the target model in lateral, isometric, and top views.

**Figure 15 materials-16-01039-f015:**
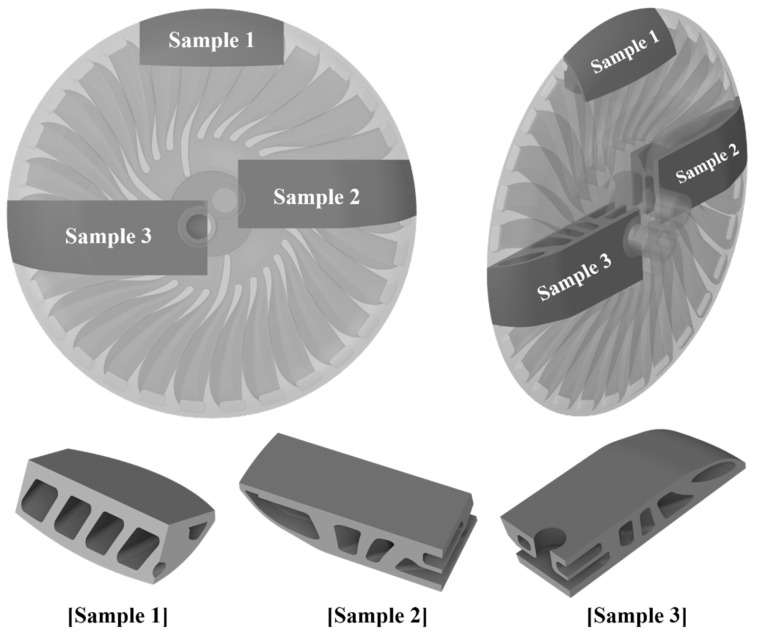
Original location and domain of extracted sample parts from the entire heat exchanger domain.

**Figure 16 materials-16-01039-f016:**
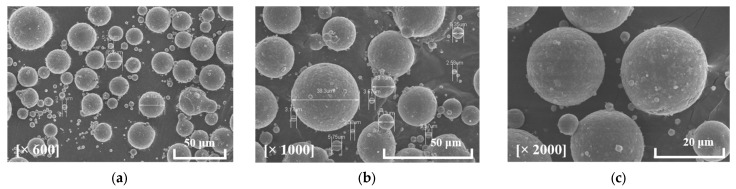
SEM images of AlSi7Mg metal powder in various magnifications: (**a**) 600×; (**b**) 1000×; (**c**) 2000×.

**Figure 17 materials-16-01039-f017:**
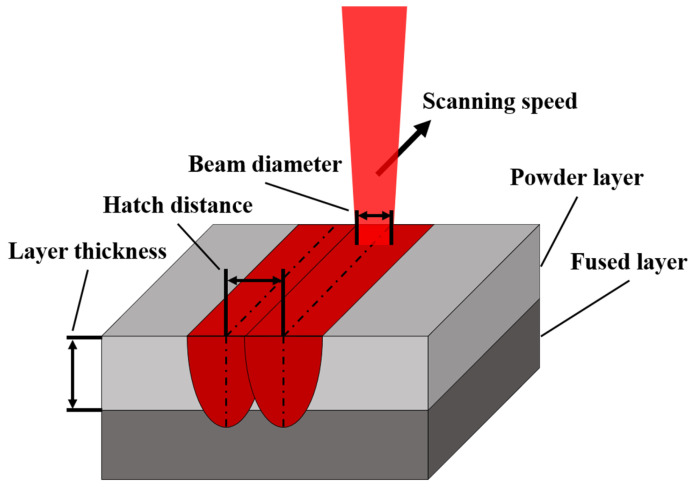
Operation parameters of the LPBF process.

**Figure 18 materials-16-01039-f018:**
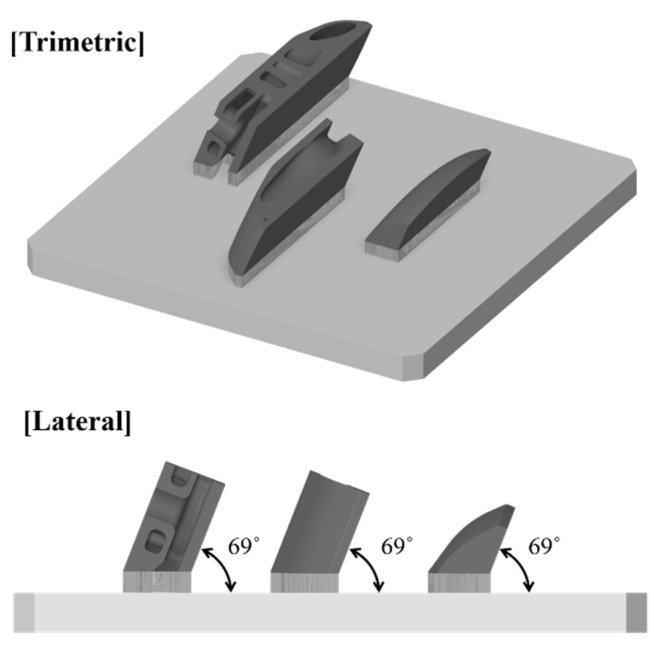
Build orientation of the samples.

**Figure 19 materials-16-01039-f019:**
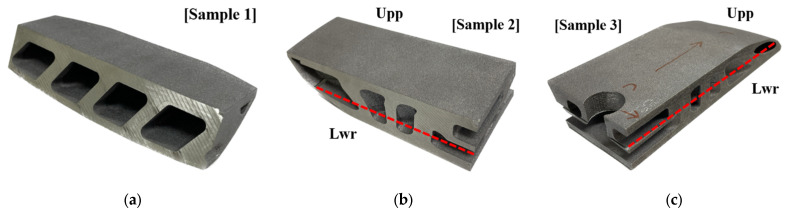
Fabricated samples corresponding to extracted samples. Samples were fabricated using AlSi7Mg metal powder via LPBF: (**a**) sample 1; (**b**) sample 2; (**c**) sample 3.

**Figure 20 materials-16-01039-f020:**
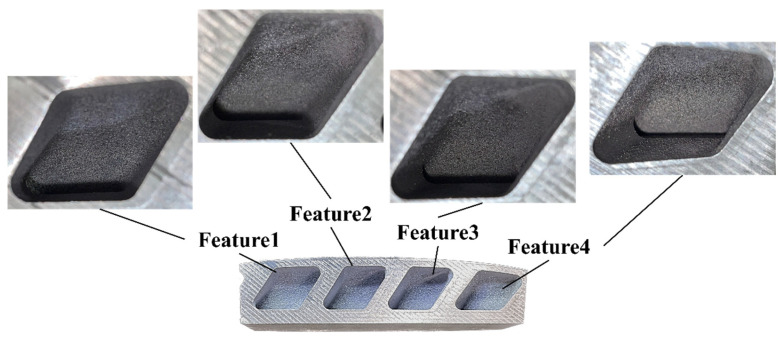
Fabricated sample 1 and its features for comparison.

**Figure 21 materials-16-01039-f021:**
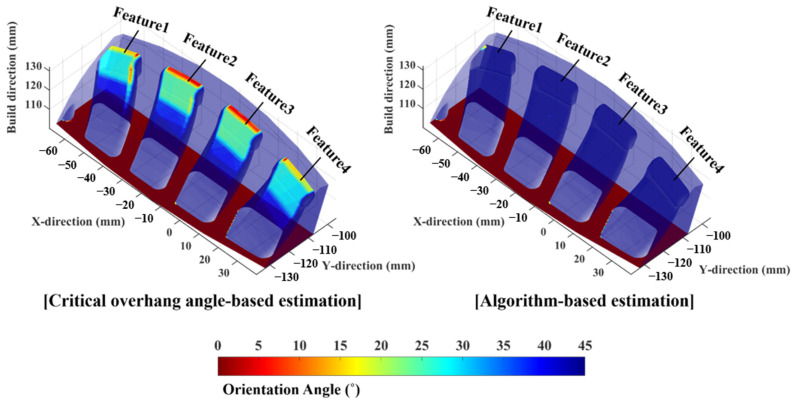
Contours of orientation angle and re-evaluated orientation angle by the algorithm over the surface of sample 1.

**Figure 22 materials-16-01039-f022:**
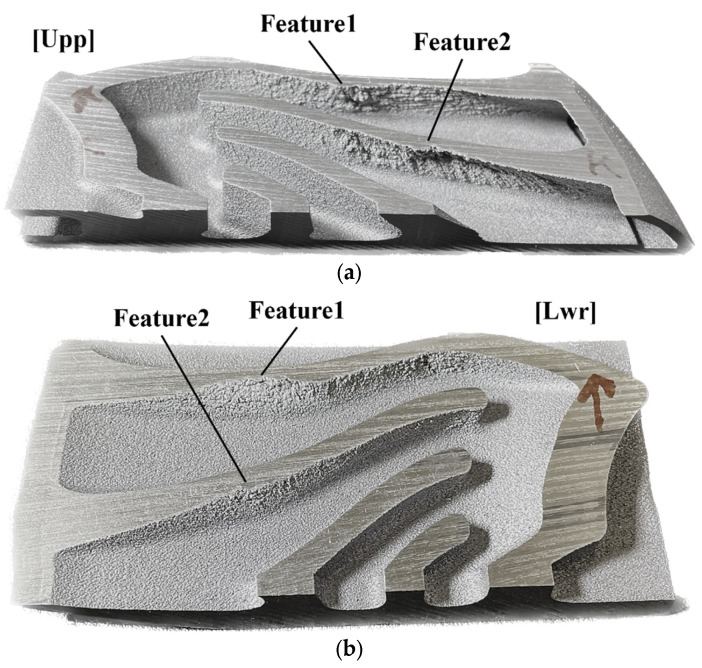
Segmented sample 2 and named features: (**a**) upper region of segmented sample 2 and two collapse regions, feature1 and feature2; (**b**) lower region of segmented sample 1 and two collapse regions, feature1 and feature2.

**Figure 23 materials-16-01039-f023:**
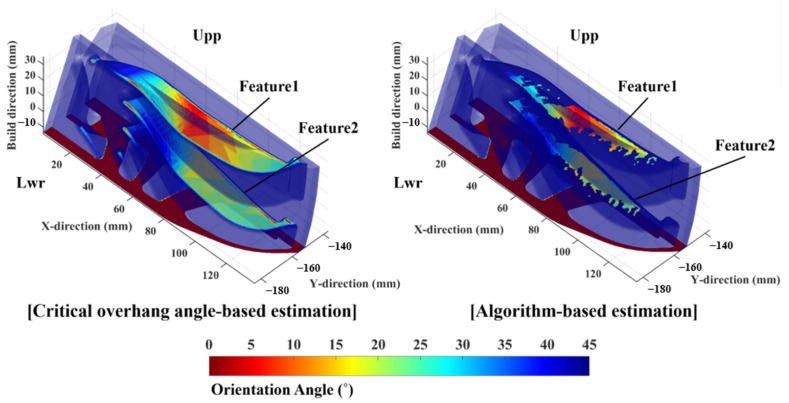
Contours of orientation angle and the re-evaluated orientation angle yielded by the proposed algorithm over the surface of sample 2.

**Figure 24 materials-16-01039-f024:**
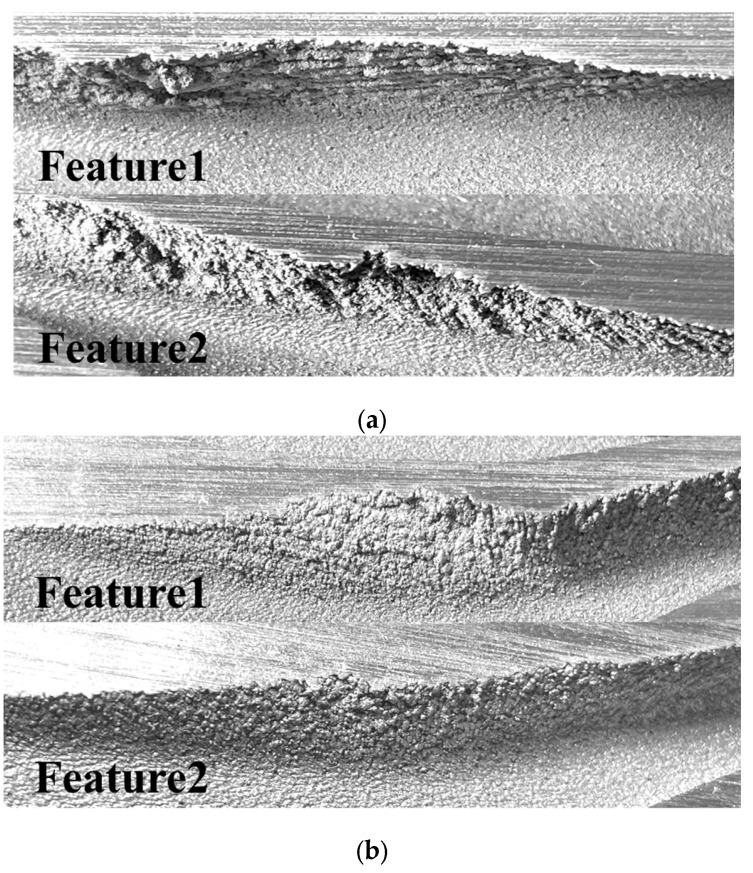
Close-up images of segmented sample 2 and named features: (**a**) upper region of segmented sample 2 and two collapse regions, feature1 and feature2; (**b**) lower region of segmented sample 2.

**Figure 25 materials-16-01039-f025:**
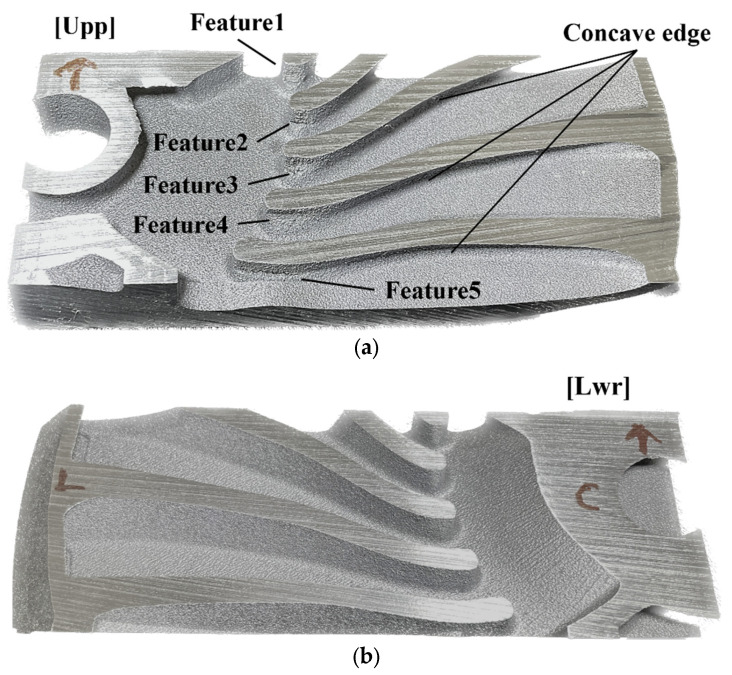
Segmented sample 3 and named features: (**a**) upper region of segmented sample 3 and collapse regions, i.e., feature1, feature2, feature3, feature4, and feature5; (**b**) lower region of segmented sample 3.

**Figure 26 materials-16-01039-f026:**
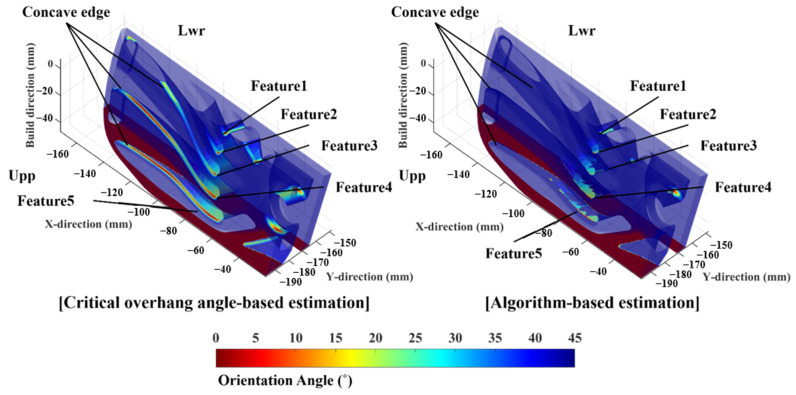
Contours of orientation angle and re-evaluated orientation angle yielded by the proposed algorithm over the surface of sample 3.

**Figure 27 materials-16-01039-f027:**
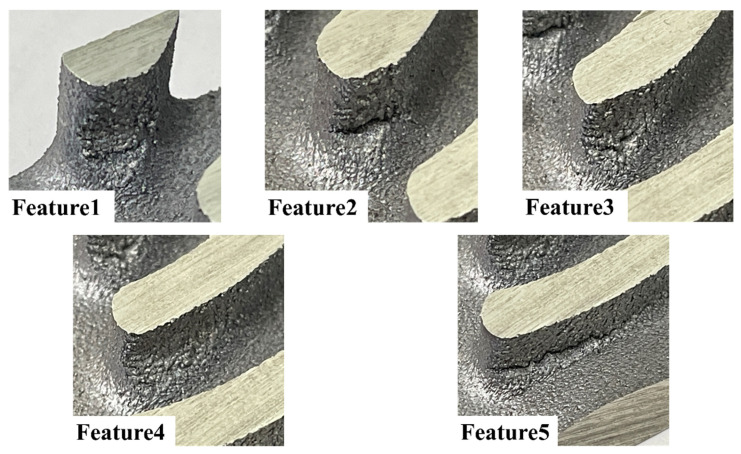
Collapse features in lower region of sample 3.

**Table 1 materials-16-01039-t001:** Thermal and physical properties of AlSi7Mg.

Properties	25 °C	90 °C
Density (g/cm^3^)	2.67	
Specific heat (J/g K)	0.913	0.901
Thermal conductivity (W/m∙K)	136.155	137.385

**Table 2 materials-16-01039-t002:** Process parameters specified for drawing infill and contour by laser.

Operation Parameter	Infill	Contour
Power (W)	370	370
Scanning speed (mm/s)	1300	1300
Beam diameter (mm)	0.11	0.075
Hatch distance (mm)	0.14	
Layer thickness (mm)	0.06	

**Table 3 materials-16-01039-t003:** Number of facets, number of iterations, and processing times for samples 1, 2, and 3.

	Sample 1	Sample 2	Sample 3
Number of facet elements	166,932	319,076	395,066
Number of iterations	101	53	92
Surface area (m^2^)	0.0189	0.0361	0.0447
Processing time (s)	441	1088	1862

## Data Availability

Not applicable.

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
