# Peer review of "Facet Connectivity-Based Estimation Algorithm for Manufacturability of Supportless Parts Fabricated via LPBF"

_materials, 2023, doi:10.3390/ma16031039_

Round 1
Reviewer 1 Report
In this paper, a novel algorithm that can detect collapse regions precisely is proposed, and a heat exchanger model with an extremely complex internal space is adopted to validate the algorithm. The proposed algorithm is helpful for predicting the surface quality, estimation of collapse estimation, and choice of build orientation for parts fabricated by Laser Power Bed Fusion. Overall, the article is well organized and its presentation is good. However, there is several issues still need to be improved.
1. In Line 409, “However, the beam diameter was 0.057 mm,” which is conflict with the Beam diameter parameter in Table 2
2. In Table 3, number of facet elements of Coupon 1 is about half the number of facets for Coupon 2 and Coupon 3. Is there any effect on the prediction results for the algorithm?
3. Coupon 1, 2 and 3 is fabricated separately rather than cut from the heat exchanger. The thermal history is different for heat exchanger and coupons. Does the difference of thermal history could cause the change of quality of the surface.
4. What is the parameters for down skin for the three Coupons, respectively?
5. The building orientation should be illustrated.
6. A parameter should be used to quantitatively characterize the degree of collapse, e.g. surface roughness.
Author Response
- The typo was corrected in the main text.
-
You have pointed out the effect of the number of the facets on part surface, and your comment is very correct. The number and size of the facets directly affects to the representation of the prediction area. However, growing number of facets dose not consistently affect to the prediction as long as the STL format describe curves and surface of the geometry enough. As you mentioned, the number of facets of coupon 1 is remarkably smaller than other facets. However, we have restricted regularization of the facets over part surface by identical maximum edge length. Thus, the difference of the facet numbers among coupons was mostly affected by the surface area of the coupons not by geometry description with facet. We have added the surface area of the coupons in the table, and the number of facets seemingly agrees with the area.
-
Your comment is right. In many cases, build quality of the surface is reciprocally interacted with the structure in upper section and lower section. In the case of former, the bad surface quality affected by the structure built above is mainly caused by heat affected zone (HAZ) when laser draws above layers. This phenomenon can be sufficiently addressed by overhang angle as suggested algorithm in this study. In the latter case, deformed or incomplete under structure may cause bad surface quality, and which was not addressed by presented algorithm. However, we have assumed that the under structure of detected collapse zones were stably and completely built. As you mentioned, building whole the part to verify collapse regions is most desirable and accurate method for verification of manufacturability. However, additive manufacturing of the large size part is extensively time and material consuming. Thus, extraction of coupons to verify manufacturability of the part is more preferred technique for large size part in the industry. Our main objective for the fabrication of the coupon was to validate prediction algorithm by demonstration of manufacturability checking process with the complex heat exchanger application. Although unexpected defection cannot be detected by the coupon extraction technique, the fabricated coupons and algorithm were addressed on the identical domains, and the algorithm showed precise prediction in the coupon domain. Thus, the prediction of the algorithm is still justified.
-
The process parameters were not changed particularly for the down skin section, and the yield angle of the down skin was mainly addressed at about 45Ëš.
-
A figure was newly added to illustrate build orientation of coupons.

Reviewer 2 Report
This is a good review article. It is definitely scientifically sound.
Some minor comments. I would like to see responses to these comments in the revised manuscript.
The title of the manuscript is not good: Algorithm for collapse estimation [?].
I would replace the word/term "coupon" for another term.
It is desirable to read about common or typical lasers used in these processes (at least a wavelength). Does it make sense to use pulsed lasers with a high frequency of pulses?
What three dots (lines 304 and 315) mean?
Are there any limitations and drawbacks of the described "algorithm"?
Author Response
- A figure was newly added to illustrate build orientation of coupons.
-
The term ‘coupon’ has been replaced by ‘sample’.
-
In the fabrication process, GE concept laser M2 series5 was adopted for manufacturing, and pulsed lasers were not applied in the process according to brochure of the machine. The machine exploits infrared laser with wavelength about 1440 nm. The brochure explains that dual lasers fuse powder with constant power. Explanation of the fabrication process was enhanced by the information of exploited machine.
-
Inappropriate uses of the symbol were eliminated.
-
According to collapsed analysis with overhang angle, the bad surface quality transferred from under structure by deformation and incomplete shaped of the structure cannot be detected. Thus, the supplementary algorithm will be addressed on the future work.

Reviewer 3 Report
The manuscript entitled “materials-2146589-LB-PBF” dealing with AM has been reviewed. The paper has been nicely written but needs significant improvement. Please follow my comments.
1. Figure 1 is a well-known fact. Please remove it.
2. Provide more discussion and fundamental relations for Figure 2. Overhang angle and support of part fabricated via additive manufacturing.
3. “Figure 6. Decision tree for detecting supporting edges within interconnection of facets”. Is this tree changing by the material type and properties?
4. What is the future direction of this work?
5. Laser absorptivity in AM is important which shows the quality of the parts and transition from keyhole to conduction mode. Please read and add the following ref in this area. “The effect of absorption ratio on meltpool features in laser-based powder bed fusion of IN718”.
6. Please update the introduction with the new publications in the field. Authors are encouraged to read and add the following new papers in the field.
· Review of quality issues and mitigation strategies for metal powder bed fusion
· Proposal of design rules for improving the accuracy of selective laser melting (SLM) manufacturing using benchmarks parts
· Fatigue life optimization for 17-4Ph steel produced by selective laser melting
Author Response
- Figure 1 was eliminated.
- An explanation of overhang angle and support was enhanced.
- The collapse of an additive manufacturing part can be changed by the materials and process conditions. The algorithm reflects different materials and process parameters by different effective length (alpha) and critical overhang angle. Thus, the decision tree is identical for different materials and properties, but effective length and critical overhang angle are changed.
- According to collapsed analysis with overhang angle, the bad surface quality transferred from under structure by deformation and incomplete shaped of the structure cannot be detected. Thus, the supplementary algorithm will be addressed on the future work.
- The reference you provided was added in the literature survey.
- The references you provided were added into the main text.
